# Interplay Between SNX27 and DAG Metabolism in the Control of Trafficking and Signaling at the IS

**DOI:** 10.3390/ijms21124254

**Published:** 2020-06-15

**Authors:** Natalia González-Mancha, Isabel Mérida

**Affiliations:** Department of Immunology and Oncology, Centro Nacional de Biotecnología (CNB-CSIC), Darwin 3, Campus UAM Cantoblanco, 28049 Madrid, Spain; ngonzalez@cnb.csic.es

**Keywords:** diacylglycerol, diacylglycerol kinase, SNX27, retromer, immune synapse, intracellular trafficking

## Abstract

Recognition of antigens displayed on the surface of an antigen-presenting cell (APC) by T-cell receptors (TCR) of a T lymphocyte leads to the formation of a specialized contact between both cells named the immune synapse (IS). This highly organized structure ensures cell–cell communication and sustained T-cell activation. An essential lipid regulating T-cell activation is diacylglycerol (DAG), which accumulates at the cell–cell interface and mediates recruitment and activation of proteins involved in signaling and polarization. Formation of the IS requires rearrangement of the cytoskeleton, translocation of the microtubule-organizing center (MTOC) and vesicular compartments, and reorganization of signaling and adhesion molecules within the cell–cell junction. Among the multiple players involved in this polarized intracellular trafficking, we find sorting nexin 27 (SNX27). This protein translocates to the T cell–APC interface upon TCR activation, and it is suggested to facilitate the transport of cargoes toward this structure. Furthermore, its interaction with diacylglycerol kinase ζ (DGKζ), a negative regulator of DAG, sustains the precise modulation of this lipid and, thus, facilitates IS organization and signaling. Here, we review the role of SNX27, DAG metabolism, and their interplay in the control of T-cell activation and establishment of the IS.

## 1. Introduction

The immune synapse (IS) consists on a highly organized, dynamic macromolecular structure that enables cell–cell communication between immune cells. The formation of this tight cellular contact between antigen-presenting cells (APCs) and T cells is extensively studied. Nevertheless, this structure is also formed in other immune events, for example, between B lymphocytes and APCs [1] or during the engagement of cytotoxic T lymphocytes (CTL) or natural killer (NK) cells with infected/tumor cells for their clearance [2,3]. T-cell receptor (TCR) recognition of an antigen on an APC leads to remodeling of the actin and microtubule cytoskeletons, resulting in an evident change in T-cell morphology and the initial formation of the IS. Under these conditions, the microtubule-organizing center (MTOC) translocates to the T cell–APC contact surface, followed by the Golgi apparatus, endoplasmic reticulum, mitochondrial network, and endosomal compartments [4,5,6].

Vesicular trafficking to the IS plays a crucial role in IS assembly and function [7]; upon T-cell activation, signaling molecules organized in microclusters, as well as scaffold and adhesion proteins, traffic toward the IS and segregate in discrete supramolecular activation clusters (SMACs). The differential distribution of these molecules facilitates the spatio-temporal regulation of downstream signaling pathways, as well as the T-cell’s effector functions [8,9,10,11]. Intracellular trafficking also mediates the polarized secretion of cytokines, lytic granules, and other cargoes toward the APC, regulating cell–cell communication [12,13,14,15,16]. Endosomal recycling represents the main mechanism via which T lymphocytes sustain continuous expression of receptors and signaling components at the IS. Sorting of internalized cargoes from early endosomes to recycling endosomes and Golgi favors polarized trafficking at the cell–cell interface in a mechanism that involves multiple proteins and phospholipids [17]. The evolutionary conserved endosomal retromer complex, in cooperation with several sorting nexin (SNX) proteins, such as SNX27, rescues transmembrane proteins from degradation and regulates their recycling [18]. The interaction of the retromer with the Wiskott–Aldrich syndrome protein and SCAR homolog complex (WASH) facilitates the nucleation of F-actin, promoting retrograde transport from early endosomes [19]. WASH deficiency in T lymphocytes results in impaired TCR trafficking and signaling defects [20]. Nevertheless, not all cargoes require the retromer for their recycling and SNX alone or additional machineries such as the retriever, which uses SNX17 for cargo recognition, or CCC complexes facilitate this process [21,22,23,24]. Although not too much is known about the retriever in IS formation, SNX17 is found with the TCR at the IS, and its silencing limits TCR and lymphocyte function-associated antigen 1 (LFA-1) expression at the cell surface, affecting IS formation and T-cell activation [25]. Contrary to their recycling, cargoes can also be transported to lysosomes for their degradation, which is achieved via the activity of endosomal sorting complexes required for transport (ESCRT) [26,27]. Phospholipids are important players of membrane trafficking, as they control physical features of membranes directly, through the recruitment of proteins or by regulating cytoskeleton-associated molecules [28,29]. Particular phospholipids named phosphoinositides (PI) work together with Rab GTPases, regulatory proteins that recruit effectors involved in the formation of vesicles, as well as their traffic and fusion. Furthermore, both types of molecules define endosomal and organelle identity [30,31,32]. When PI, Rab GTPases, and their effector proteins bring membranes in close proximity, soluble NSF (*N*-ethylmaleimide-sensitive factor) attachment protein receptor (SNARE) proteins go into action and mediate their fusion [33,34].

All in all, intracellular trafficking is essential for establishment of the IS and maintenance of its associated signaling. The distinct trafficking steps acting together to regulate this process involve multiple players. Here, we summarize the current knowledge of the roles of lipids, with a special focus on the lipid second messenger diacylglycerol (DAG), as well as that of the transport protein SNX27, in the spatio-temporal regulation of trafficking and signaling that sustains the IS.

## 2. Diacylglycerol and Phosphatidic Acid in the Regulation of the Immune Synapse

DAG is a lipid with critical functions in lipid metabolism and signaling. Recognition of antigens by T lymphocytes results in the rapid generation of DAG that is maintained and restricted to the cell–cell interface, facilitating the recruitment and modulation of proteins involved in T-cell polarization, immune synapse formation, and signaling [35]. Upon T cell–APC engagement phospholipase C γ1 (PLC-γ1)-mediated hydrolysis of phosphatidylinositol (PtdIns) (4,5)-bisphosphate (PtdIns(4,5)P_2_) leads to the production of inositol triphosphate (IP_3_) and DAG [36]. IP_3_ triggers the release of stored intracellular Ca^2+^ and activation of nuclear factor of activated T cells (NFAT)-mediated transcription [37,38]. DAG generation facilitates IS recruitment and activation of proteins containing DAG-binding type 1 PKC (C1) domains such as guanyl nucleotide-releasing protein for Ras1 (RasGRP1) and protein kinase C α (PKCα), that link the DAG generated at the IS to the intensity of Ras/ERK activation and activator protein 1 (AP-1)-dependent transcription [39,40,41,42]. DAG also binds and activates other PKC isoforms, like PKCθ, connecting DAG production to the regulation of the nuclear factor κB (NF-κB) pathway [43], as well as to the activation of the PDK-1/AKT/mTOR axis [44], which is involved in the regulation of protein synthesis, cellular metabolism, and cell survival. DAG levels are tightly regulated by diacylglycerol kinases (DGKs), which phosphorylate it into phosphatidic acid (PA), limiting recruitment of DAG effectors and the activation of downstream signaling pathways [35] (Figure 1A). In T lymphocytes, the main DGK isoforms contributing to DAG attenuation are type I DGKα and type IV DGKζ [45,46].

One of the characteristic events in IS formation is the translocation of the MTOC to the T cell–APC contact area. Although the exact mechanisms governing the dynamics of this event are not completely understood, DAG accumulation at the IS is followed by MTOC translocation, and it is sufficient to induce this process independent of the TCR. Inhibition of PLC-γ1 activity impairs MTOC translocation [47], whereas sequential DAG-dependent recruitment of PKCε, η, and θ facilitates MTOC localization to the IS. Indeed, small interfering RNA (siRNA)-silencing of PKCθ alone or the combination of PKCε + PKCη impaired this process [48]. Although the precise procedure via which the distinct PKC isoforms contribute to MTOC translocation is not yet known, their role in regulating dynein and myosin motors is likely involved [49]. Interestingly, failure to form a stable DAG gradient by DGK inhibition or addition of the DAG analogue phorbol-12-myristate-13-acetate (PMA) impaired both dynein recruitment and MTOC translocation [47,49].

Remarkably, DAG accumulation influences other aspects of the IS structure and signaling. For instance, dynein motors recruited to DAG-enriched areas associate with TCR microclusters and mediate their transport toward the central region of the IS [50,51]. Furthermore, these motors facilitate the localization of integrins in the peripheral area, providing an adhesive ring that stabilizes the T cell–APC conjugate [52]. DAG also drives the activation and polarization of the serine/threonine kinase PKD (protein kinase D, also known as protein kinase Cη (PKCη)) to the IS [53]. Activated PKD was shown to regulate the activity and clustering of β1 integrin induced by T-cell stimulation [54], cell proliferation, and TCR signaling via crosstalk with the DAG–PKC axis [55]. Moreover, PKD is involved in the maturation, polarized transport, and secretion of multivesicular bodies toward the IS [56]. Furthermore, other C1-containing proteins like mammalian unc (Munc) proteins, which regulate neurotransmitter release in the brain [57,58,59,60], participate in the control of lytic granule secretion by CTL and NK cells, and their defect is associated with human immunodeficiencies [61,62,63,64,65]. These proteins likely mediate exocytosis at the synapse by facilitating assembly of SNARE complexes required for vesicle fusion.

PA-mediated functions at the IS remain largely unknown. This DAG metabolite recruits phosphatidylinositol 4-phosphate 5-kinase (PIP5K) to the plasma membrane, promoting PtdIns(4,5)P_2_ generation [66,67,68]. A recent study showed that PA is evenly distributed across the plasma membrane of CTL and appears to remain unchanged during IS formation with target cells [69]. Additionally, it was revealed that the recruited PIP5K regulates the actin cytoskeleton at the IS, facilitating targeted granule secretion by the CTL. PA-regulated exocytosis was previously described in a variety of cell types including neutrophils and neurons [70,71], and it was mentioned to be facilitated by PA’s conical shape, which induces negative membrane curvatures, as well as its participation in the modulation of SNARE complexes [72,73,74]. These data suggest that PA, as DAG, may be influencing membrane IS structure and function.

All in all, these studies highlight the importance of the spatio-temporal regulation of DAG metabolism in the recruitment of effector proteins and generation of PA, which in turn influence immune synapse signaling and structure (Figure 1B).

## 3. Diacylglycerol Kinase ζ Modulates Diacylglycerol Abundance and Associated Signaling at the Immune Synapse

As mentioned in the previous section, DGK catalyzes the phosphorylation of DAG to PA, limiting DAG-regulated functions [35]. The 10 mammalian DGK isoforms described to date are classified into five groups (I–V) based on the presence of specific regulatory domains within their sequences. All isoforms include a catalytic domain and have at least two conserved C1 domains. Nevertheless, only some of these C1 domains enclose the necessary residues to bind DAG, and the mechanism of interaction with this lipid remains to be elucidated [75]. Type I DGKα and type IV DGKζ are the two isoforms expressed in T lymphocytes [45,46]. As observed for DAG, these kinases translocate to the IS following TCR/CD28 engagement and contribute to regulate the levels of this lipid [76,77]. This ensures an adequate regulation of TCR signal intensity and duration.

Despite displaying overlapping intracellular localizations and redundant roles, DGKζ was shown to exert a stronger negative function over DGKα in activated T cells by limiting PKCθ/PDK-1 mutual activation. This provides negative regulation not only of the NFκB axis, but also of the PDK-1/AKT/mTOR/S6K pathway [78]. Furthermore, this DGK isoform has a predominant role in terminating the RasGRP1/Ras/ERK pathway downstream of the TCR [46,79,80,81]. Indeed, our group showed that DGKζ controls DAG metabolism at the IS and negatively regulates PKCα translocation, a DAG effector involved in Ras/ERK activation [42,76]. Among the specialized functions of DGKζ we also found that it limits cytokine-mediated expansion of innate-like CTL independent of antigen stimulation [82]. All these findings correlate with studies in DGKζ-deficient mice presenting enhanced anti-tumoral responses, which are not observed in DGKα-deficient mice [82,83,84].

In agreement with the role of DGKα and DGKζ in the modulation of DAG at the IS, several studies showed substantial defects in IS organization as a result of DGKα or DGKζ deficiency. For instance, stimulated CD4^+^ T cells treated with DGK inhibitors or deficient for DGKα present destabilized synaptic DAG accumulation and impaired MTOC recruitment [47,77]. Moreover, DGKζ-deficient CTL show an impairment in MTOC docking to the IS, which correlates with a reduced translocation of phosphorylated PKCζ [85]. PKCζ is known to be regulated by DGK-derived PA in non-T cells [86], and it was shown to promote MTOC polarization in primary CD4^+^ and CD8^+^ T cells [87,88], suggesting a role for the DGKζ/PA/PKCζ axis in this process. Remarkably, the impact of this kinase is not limited to T cells, as DGKζ-mediated PA production was shown to play a role in the assembly of the B-cell IS, regulating actin remodeling, MTOC translocation, force generation, and antigen-uptake related processes [89].

## 4. Sorting Nexins in Membrane Trafficking

### 4.1. SNX-FERM and SNX-BAR Subfamilies

The sorting nexin (SNX) family is composed of proteins involved in the regulation of intracellular trafficking and endosomal signaling [18,90,91]. They are characterized by the presence of a phox homology (PX) domain, which is involved in phosphoinositide binding. Thus, it targets SNX to endosomal membranes, most commonly by binding to PtdIns(3)P [92,93]. Additionally, accumulating evidence demonstrates the PX domain’s ability to engage in protein–protein interactions, although its functional importance in SNX is still not clear [93]. To date, 33 mammalian SNX proteins were identified, and they are divided into subfamilies based on the presence of different structural domains. Given their relevance in membrane trafficking, we proceed to describe SNX-BAR (Bin, amphiphysin, Rvs) and SNX-FERM (4.1/ezrin/radixin/moesin) subfamilies.

SNX17, SNX31, and SNX27 form the SNX-FERM subfamily, which is characterized by the presence of an atypical C-terminal FERM domain, named the FERM-like domain. Comparable to the typical one, it contains F1, F2, and F3 modules, although the F2 sequence is shorter [92]. The F1 module resembles a Ras-association domain (RA) to which small GTPases from the Ras-like protein family bind. Additional contacts of these molecules with the F2 module may contribute to binding specificity. In vitro, all SNX FERM domains bind to H-Ras, although cell biology studies later functionally linked the SNX27 FERM domain to K-Ras [92,94,95]. The existence of additional Ras isoforms interacting with SNX FERM domains in vivo remains to be examined. Peptide array screening revealed that the F3 module specifically interacts with cargoes containing NpxY/NxxY motifs with a preference for sequences phosphorylated at Y_0_ in the case of SNX27. Of note, in vivo studies only demonstrated a role for SNX17 and SNX31 in the recycling of NpxY/NxxY-containing transmembrane proteins, preventing their lysosomal degradation [21,25,96,97,98,99,100]. All in all, these data indicate that SNX-FERM proteins are involved in endosomal cargo recycling and act as scaffolds for signaling complexes.

SNX-BAR proteins are defined by the presence of a BAR domain and include SNX1–SNX9, SNX18, SNX30, SNX32, and SNX33 [93]. The lipid-binding BAR domain is a protein dimerization motif which senses and binds positive membrane curvatures, promoting their tubulation [101,102,103]. Furthermore, members of this subfamily form part of the retromer complex and participate in intracellular cargo trafficking [103,104,105]. The retromer complex, firstly identified in *Saccharomyces cerevisiae*, is formed by a cargo-selection subcomplex (CSC) and a membrane-deforming subcomplex [106]. In mammals, the CSC is conserved and is formed by Vps35, Vps29, and Vps26A/Vps26B proteins [107,108,109], while the membrane-deforming subcomplex includes the SNX-BAR heterodimer of SNX1/SNX2 with SNX5/SNX6/SNX32 [106]. Nevertheless, the CSC was also described to bind the non-BAR domain containing SNX3, as well as SNX27, leading to different retromer structures [91]. The retromer plays a key role in the regulation of endosome-to-*trans*-Golgi transport and endosome-to-plasma membrane recycling, preventing cargo degradation [110,111,112]. This tubular-based endosomal sorting is facilitated by retromer´s association to cytoskeleton components such as the motor dynein/dynactin complex or the WASH complex, which is involved in promoting actin polymerization [113,114,115].

### 4.2. The SNX27–Retromer Multiprotein Complex Is Involved in Protein Recycling

Like all members of the SNX-FERM subfamily, SNX27 contains a PX domain and a FERM-like domain. Additionally, it includes an N-terminal postsynaptic density 95/discs large/zonula occludens-1 (PDZ) domain which makes it unique within the SNX. The PDZ domain simultaneously binds to PDZ-binding motif-containing proteins and the Vps26 subunit of the retromer complex, which enhances cargo binding affinity and favors their recycling [114,116]. The PX domain mediates SNX27 localization to PtdIns(3)P-enriched endosomal membranes. Meanwhile, the FERM-like domain can recruit cargoes containing NpxY/NxxY motifs in vitro, engage K-Ras in a GTP-dependent manner [92,95], and bind bi- and tri-phosphorylated PI [117]. Moreover, this domain interacts with SNX-BAR proteins and the WASH complex [114,115], which in turn maintains the integrity of the endosomal and lysosomal network by regulating actin polymerization [19,118,119]. SNX-BARs and the WASH complex also indirectly recruit SNX27 to the retromer and situate it as a core component of the SNX27–retromer multiprotein complex [120,121] (Figure 2A).

In this situation, SNX27 acts as an adaptor protein which links cargo recognition through its PDZ domain and likely through the FERM-like domain to retromer-mediated endosomal transport. This allows cargoes recycling to the plasma membrane, preventing their sorting into the lysosomal degradative pathway [112,114,115] (Figure 2B,C). Some examples include the β2-adrenergic receptor [115,122], the glucose transporter 1 (GLUT1) [114], the G-protein-gated potassium (Kir3) channel [123], or the glutamine transporter (alanine, serine, cysteine transporter 2 (ASCT2)) [124].

## 5. SNX27 in the Regulation of the Immune Synapse

Our laboratory identified SNX27 expression in T lymphocytes in a proteomic analysis that searched for PDZ-interacting DGKζ partners [125]. In resting T lymphocytes, SNX27 localizes at early and recycling endosomes mainly through the interaction of its PX domain with PtdIns(3)P [125]. Upon antigen presentation by an APC, these SNX27-enriched compartments rapidly polarize toward the cell–cell interface with an important SNX27 fraction accumulating at the central and peripheral SMAC (c-SMAC and p-SMAC) of the IS (Figure 3). This polarized trafficking is mediated by the binding of the PX domain to PtdIns(3)P and the FERM domain to PtdIns(4,5)P_2_- and/or PtdIns(3,4,5)P_3_-enriched membrane regions [117,126]. The SNX27 PDZ domain also influences this process, although the specific PDZ-binding motif-containing cargoes directing SNX27 recruitment to the IS still remain unknown [126,127].

The participation of SNX27–retromer in intracellular trafficking and its accumulation at the IS suggest a role in the transport of cargo to the cell–cell interface. Proteomic analysis of the SNX27 interactome from IS-forming T cells confirmed its association with DGKζ, the retromer and WASH complexes, and additional cargoes that associate to SNX27 to traffic to the IS [127]. These include the protein zonula occludens-2 (ZO-2), a constituent of tight junctions never before identified in T lymphocytes, centromere protein J (CENPJ), which is part of the centrosome, or the Rho guanine nucleotide exchange factor 7 (ARHG7, also known as β p21-activated kinase-interactive exchange factor (β-PIX)) among others. ZO-2 mobility at the IS was decreased in SNX27-silenced T cells, consistent with the idea that SNX27 coordinates polarized trafficking toward this structure (Figure 3). Unlike that observed in other cell systems, PDZ-interacting SNX27 cargoes during IS formation were proteins with functions in cytoskeletal remodeling, cell adhesion, and/or centrosome organization, suggesting that SNX27 functions as a signaling scaffold in T cells, likely constituting an important regulator of activation at the IS. This role is facilitated by its specific structural composition; the F1 module of the FERM domain binds to GTPases from the Ras-like protein family [92,95], while the PDZ domain interacts with scaffolds and signaling complexes. For instance, it engages cytohesin-associated scaffolding protein (CASP), which regulates signaling through the ADP-ribosylation factor (ARF) family of small GTPases [128] or Kidins220, a scaffold that promotes ERK signaling [129].

Immune synapses not only resemble neuronal synapses in morphology, but they also share common transport mechanisms. Indeed, release of secretory vesicles at the IS is highly similar to neurotransmitter delivery from neurons and neuroendocrine cells [130]. Remarkably, SNX27 is also present in neurons, where it was described to play a key role facilitating PDZ-mediated recycling of cargoes, such as glutamate receptors [95,131]. The importance of this complex is underscored by the fact that SNX27 deficiency or mutation is associated with synaptic dysfunction and a variety of neurological diseases such as Alzheimer’s disease or Down syndrome [131,132]. Additionally, recent studies suggested that SNX27–retromer-mediated trafficking favors breast cancer metastasis [133,134]. Therefore, further research on SNX27 will shed light on immune and neuronal function and dysfunction, as well as increase the understanding of tumor invasiveness.

## 6. SNX27 Interacts with DGKζ and Modulates Diacylglycerol Metabolism at the Immune Synapse

Structural studies of PDZ-based SNX27 and cargo interaction demonstrated that this binding can be of high or low affinity based on the amino-acid sequence upstream of their PDZ-binding motif. High-affinity engagement to SNX27 requires cargoes containing acidic residues located at the -3 and -5 positions, which are able to clamp a conserved arginine on the SNX27 surface. Nevertheless, cargoes that lack these acidic residues, but instead present conserved phosphorylation sites, can also undergo high-affinity binding to SNX27 upon phosphorylation of these residues [135]. In agreement with our identification of SNX27 association with DGKζ in T cells [125], biophysical and biochemical analyses confirmed that the DGKζ terminal sequence EDQETAV promotes a high-affinity interaction with the SNX27 PDZ domain [135] (Figure 4A).

The best-known function of SNX27–retromer is to promote recycling of PDZ-binding transmembrane cargoes, preventing their degradation. Consequently, SNX27 silencing is reported to enhance degradation of many of its binding partners, as observed for GLUT1 [114]. However, we did not detect significant changes in DGKζ protein levels in SNX27-silenced cells, suggesting that this interaction is not required to maintain DGKζ expression but its spatial localization [136].

The identification of a PDZ-mediated interaction of SNX27 with DGKζ and the finding that these proteins accumulate at the IS following T-cell stimulation prompted us to study the participation of SNX27 in the modulation of DAG-regulated pathways upon TCR antigen recognition (Figure 4B). Research from our group revealed that silencing of either SNX27 or DGKζ in antigen-stimulated T cells results in increased ERK phosphorylation, suggesting a functional connection between these two proteins [126]. Moreover, SNX27 silencing results in NF-κB hyperactivation upon TCR co-stimulation that does not further increase following additional DGKζ downregulation [78,136]. Although these data suggest redundant functions for SNX27 and DGKζ in T-cell signaling, it is worth mentioning that SNX27 may modulate T-cell activation via interaction with additional cargoes. For instance, TCR co-stimulation of DGKζ-silenced cells promotes PKCθ interaction with PDK1 and subsequent mTOR signaling activation [78], while SNX27 silencing downregulates this pathway [136]. The decreased mTOR activation in SNX27-silenced cells correlated with defective antigen-induced growth of naïve T cells from *Snx27^−/−^* mice. This raises the possibilities that SNX27 silencing either potentiates DGKζ inhibitory function on the mTOR pathway or disrupts interaction of SNX27 with cargoes that favor mTOR activation independent of DGKζ.

In summary, the identification of SNX27 as a DGKζ-interacting partner that facilitates its strict spatial and temporal regulation during IS formation offers new insight into the precise modulation of DAG metabolism and signaling during T-cell activation (Figure 4B). Adequate equilibrium of DAG and PA not only favors regulation of signaling molecules, but it also modulates proteins involved in polarization and intracellular transport, and it contributes to inducing negative membrane curvatures, important for membrane fission and fusion [72]. Further research will be needed to determine the extent to which SNX27 regulation of DAG metabolism impacts IS formation and efficiency.

## 7. Concluding Remarks

Precise regulation of intracellular transport is particularly crucial in polarized cells, which depend on active membrane trafficking at specific sites to carry out their functions [137,138,139]. In T cells, polarized membrane trafficking facilitates T-cell surveillance, surface display of receptors, as well as adhesion and signaling molecules, and release of cytokines and other cargo to the immunological synapse [7,137]. Indeed, loss of polarity is associated with impaired signaling competence. Numerous players are involved in the regulation of these processes, with a remarkable participation of lipids.

The findings summarized in this review highlight the important contribution of SNX27 in the regulation of DAG metabolism and the influence of these molecules in intracellular trafficking and signaling for establishment and maintenance of the IS. Although we only offer a view of the T-cell side of the IS, localized trafficking also occurs at the APC side and greatly influences the stability and activity of this structure [140,141]. Thus, it will be of great interest to investigate if these molecules also play a role in the polarization and signaling triggered in the APC. Furthermore, some findings on SNX27 and DAG dynamics during IS formation could be extrapolated to other models of polarized trafficking such as invadopodia formation or the neuronal synapse, where SNX27 is associated with numerous pathological conditions [131,132,133,134].

## Figures and Tables

**Figure 1 ijms-21-04254-f001:**
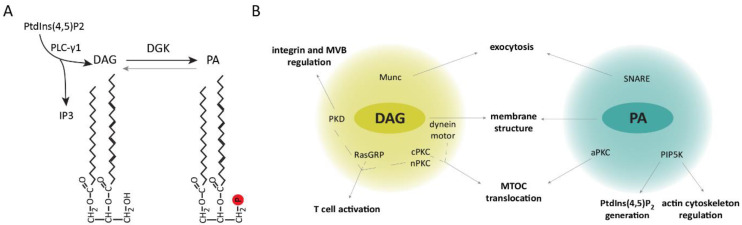
Diacylglycerol (DAG) metabolism contributes to immune synapse (IS) structure and associated signaling. (**A**) Upon T-cell activation, PLC--γ1-mediated phosphatidylinositol (4,5)-bisphosphate (PtdIns(4,5)P_2_) hydrolysis leads to the generation of inositol triphosphate (IP_3_) and DAG, which in turn can be converted to phosphatidic acid (PA) via the activity of a diacylglycerol kinase (DGK). (**B**) Schematic representation of proteins recruited and modulated by DAG and PA at the synapsis of T lymphocytes, indicating the cellular processes in which they are involved. MVB, multivesicular bodies; nPKC, novel protein kinase C (PKC); cPKC, conventional PKC; aPKC, atypical PKC.

**Figure 2 ijms-21-04254-f002:**
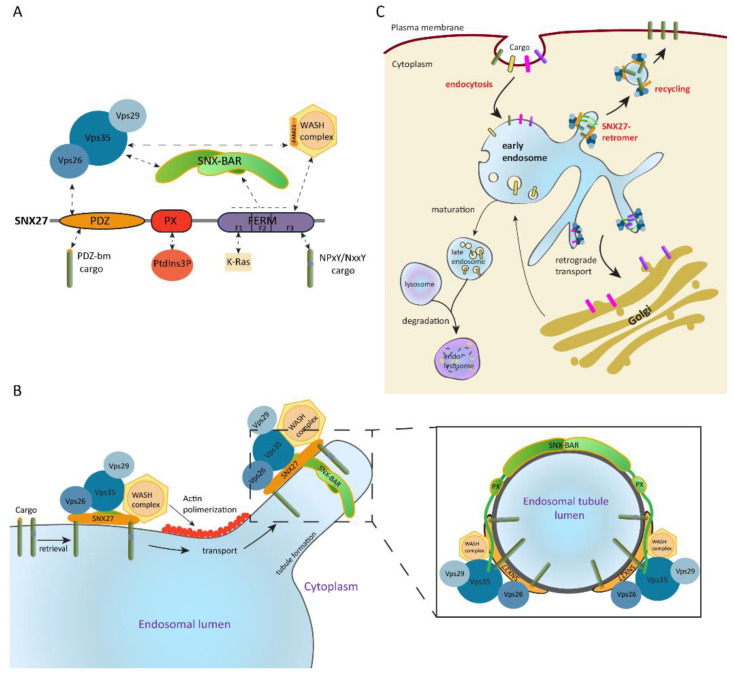
Sorting nexin 27 (SNX27)–retromer architecture and function in intracellular trafficking. (**A**) SNX27 establishes N-terminal postsynaptic density 95/discs large/zonula occludens-1 (PDZ) domain-mediated interactions with cargo and the Vps26 retromer subunit, as well as 4.1/ezrin/radixin/moesin (FERM) domain-dependent engagement with the retromer SNX-BAR (Bin, amphiphysin, Rvs) and the Wiskott–Aldrich syndrome protein and SCAR homolog (WASH) complex. Moreover, both the SNX-BAR dimer and the WASH complex directly bind to the cargo selection subcomplex of the retromer. (**B**) SNX27–retromer promotes endosomal trafficking of cargo to the plasma membrane; this multiprotein complex is associated to the cytosolic face of the endosomal membrane mainly through binding of the SNX27 phox homology (PX) domain and SNX-BARs to phosphoinositides (PI). Endosomal localization is further stabilized by SNX27 cargo recognition and cargo-selection subcomplex (CSC) interaction with SNX-BAR and SNX27. Actin polymerization mediated by the WASH complex and membrane remodeling induced by SNX-BAR mediate tubule formation and scission of the cargo-enriched endosome subdomain. A frontal view of the endosomal tubule coated by the SNX27–retromer is shown. Images modified from References [17,112]. (**C**) The generated cargo-loaded vesicles are subsequently recycled to the cell surface, preventing their lysosomal degradation. Image modified from Reference [111].

**Figure 3 ijms-21-04254-f003:**
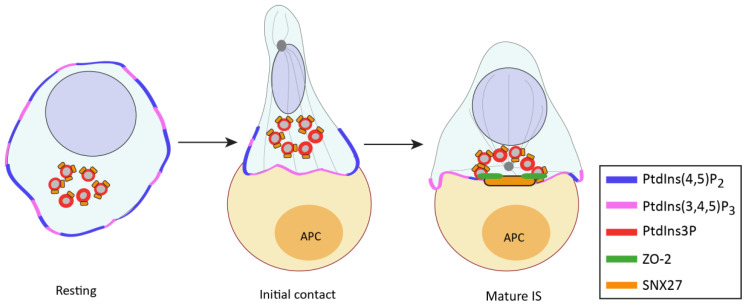
SNX27 is recruited to the IS and facilitates trafficking of cargoes toward this structure. In resting T cells, SNX27 is mainly found at PtdIns(3)P-positive endosomes. Upon T-cell activation, these endosomes polarize toward the cell–cell interface, and a fraction of this protein accumulates at the IS. This event is facilitated by the SNX27 PDZ domain and the interaction of its FERM domain with PtdIns(4,5)P_2_- and/or PtdIns(3,4,5)P_3_-enriched membrane regions. Binding of SNX27 to its cargoes can drive their mobility to the IS, as observed for zonula occludens-2 (ZO-2).

**Figure 4 ijms-21-04254-f004:**
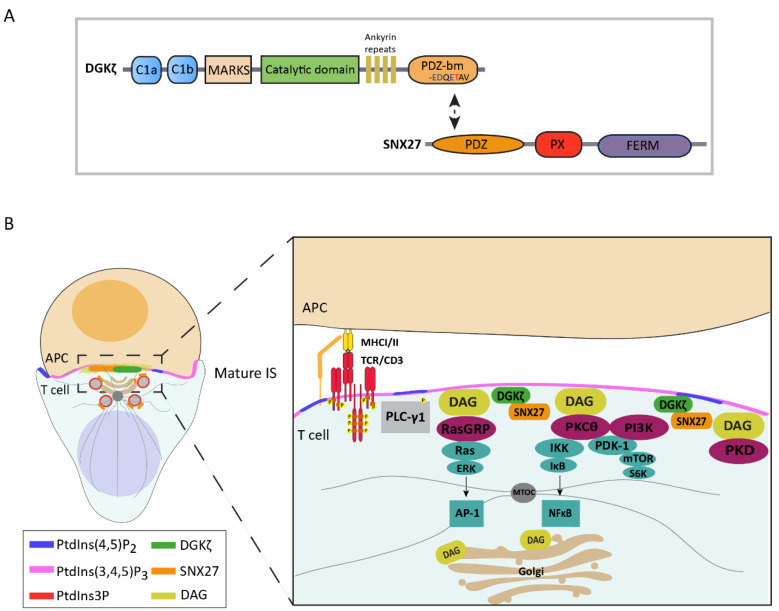
SNX27 participates in the regulation of DAG metabolism at the IS through interaction with DGKζ. (**A**) DGKζ and SNX27 proteins interact in a PDZ-dependent manner. The high-affinity DGKζ C-terminal sequence EDQETAV with possible phosphorylation sites (red) and positively charged amino acids (blue) is indicated. (**B**) Engagement of antigen–TCR triggers PLC-γ1 activation, resulting in DAG production and its accumulation at the IS. Likewise, SNX27 and the DAG-negative regulator DGKζ translocate to the IS simultaneously with PtdIns(3,4,5)P_3_ and DAG generation. SNX27 sustains the stability and localization of this kinase, facilitating the regulation of DAG and its downstream signaling pathways.

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
