# Peer review of "Interplay Between SNX27 and DAG Metabolism in the Control of Trafficking and Signaling at the IS"

_ijms, 2020, doi:10.3390/ijms21124254_

Round 1

Reviewer 1 Report

This is an interesting and timely review with the originality of being in part focus on T cells and the immunological synapse.

Overall it is very clear and describes very well this field.

I just have a few comments/suggestions for the authors that would improve the manuscript.

Summary:  T cells undergo a conformational change : Conformational change is rather use for proteins. Would be better to use reorganization?

Introduction

 line 27:  CTLs also for immunological synapses with equal functions than NK cells against infected/tumor cells but in the adaptive immune response: See work & reviews by G. Griffiths

Line 36: A word on signaling microclusters should be said with the original refs, including Campi, Dustin et al that together with Yokosuka, Saito et al znd Bunnell et al 2002 were the pioneer puvblications

Line 37: Not only endosomal, but also Golgi and lysosomal compartments for the cytokines and lytic granules

2.1

Line 144: It would be interesting to cite also work by De Saint Basile et al and others on immune deficiencies, CTLs and immune synapse and immune cell secretion: i.e. J. Feldman et al 2003, Menager et al 2007, Neeft, van der Sluijs et al 2005

5.

Line 261: SNX27 fraction accumulating at the central SMAC (c-SAMC) peripheral SMAC (p-SMAC): Which one ? c-SMAC or p-SMAC, or both?

Reviewer 2 Report

Interplay between SNX27 and DAG metabolism in the control of trafficking and signaling at the immune synapse

Brief Summary

The goal of this review is to describe the role of SNX27, DAG Metabolism and their interplay during T-cell activation and immune synapse (IS) formation. Introduction provides background on the immune synapse and broad outlook at intracellular trafficking. This is followed by a thorough overview of the role of lipids in membrane trafficking. The phosphoinositide section introduces DAG as a key molecule. This serves as a transition into the next section covering DAG and its role in regulating the IS. The other player of interest is SNX27, which is introduced in more detail following DAG introduction. The review ends by combining knowledge and revealing a new regulation of DAG by SNX27 during IS. Overall review does a good job of reviewing the intended material of DAG and SNX27 at the IS but the impact is decreased due to the heavy emphasis on lipids and endosomal trafficking.

Broad comments

Strengths

  • This review provides a detailed description of the role of phosphoinositides in membrane trafficking.
  • The role of DAG in regulating the immune synapse is well described.
  • The modulation of DAG by SNX27 during IS formation is also detailed and well described. This is the strongest section, which is good as this is the main focus of the review. That being said I wonder if it could be expanded.
  • Similarly, the knowledge of DGKs is strong, almost seems like that could have its own section as well.
  • The interactions and individual domains of different proteins are also well described.
  • Most figures were clear and helpful.

Weaknesses

  • This review feels like two different mini reviews. One that focuses on phosphoinositides and their role in trafficking, and a second review that discusses the relationship between SNX27 and DAG and their role in IS.
  • The review frequently discusses endosomal trafficking but either due to wording or lack of clear explanation it is often hard to distinguish the direction of trafficking and its relation to IS formation. Some of the wording seems more analogous to the secretory pathway rather than the endocytic pathway. It remains unclear what aspects of the endocytic pathway influence the IS. See specific comments.
  • While the role of lipids and trafficking relates to the generation of DAG and the role of SNX27, the material presented is not quite put in the context of T-cell activation and IS formation and does not feel well integrated with the rest of the paper.
  • The title suggests that the interplay between SNX27 and DAG regulates trafficking but it really only seems to discuss their effects on signaling.
  • The review states DAG metabolism as being important for IS. This suggests that the metabolite may be discussed but it is unclear what role phosphatidic acid has on IS.
  • Minor but the use of introductory phrases is much too frequent.  In  some cases, the chosen introductory phrase does not fit or takes away from the sentence. This effect seems to be more common with the use of “Besides” and “on the other hand”.

Specific Comments

  • From the abstract, it is not evident that large portion of this review will discuss endosomal trafficking and phosphoinositides.
  • Paragraph 2 lines 35-39 seem to be referring both to secretory pathway processes, which is confusing as introductory line mentions endosomal trafficking.
  • Paragraph 2 of the intro seems to try to introduce endosomal trafficking but seems rather confusing. A more general paragraph where the endosomal pathway is described and its role in IS would be beneficial.
  • Phosphoinositide section mentions there are 7 distinct PI species. There is no mention of or lack of relevance of ptdins(3,4)p2 and ptdins(5)p. IP3 also not really mentioned here.
  • DAG and PA are mentioned to be involved with SNARE complexes but there is no mention of the relationship between SNARES and the immune synapse.
  • Figure 2B It might be use full to split this image into separate smaller images (to show temporal progression) rather than a before and after picture. The dash lines make it seem like its zooming into the image.
  • Could you clarify if during IS formation SNX27 plays largely a trafficking role or scaffolding role. Line 266 suggests trafficking, but the following sentences mainly mention interactors and scaffolding examples. If it does play a role in trafficking a picture of SNX27 during IS formation may also be relevant.
  • The regulation of the Ras/ERK/AP-1, 176 PKCθ/PKD, NFκB and mTOR1 metabolic pathways by DAG is briefly mentioned in lines 176-177 and 314-322. Considering their relevance to IS formation and their presence in one of the figures this section feels like it could be expanded.
